# Blood Urea/Creatinine Ratio and Mortality in Ambulatory Patients with Heart Failure with Reduced Ejection Fraction

**DOI:** 10.3390/diseases13110362

**Published:** 2025-11-07

**Authors:** Andrew S. Oswald, Muhammad S. Hussain, Mohsin H. K. Roshan, Filippo Pigazzani, Anna-Maria Choy, Faisel Khan, Ify R. Mordi, Chim C. Lang

**Affiliations:** Division Cardiovascular Medicine, School of Medicine, University of Dundee, Dundee DD1 9SY, UKm.s.z.hussain@dundee.ac.uk (M.S.H.); mroshan001@dundee.ac.uk (M.H.K.R.); f.pigazzani@dundee.ac.uk (F.P.); a.choy@dundee.ac.uk (A.-M.C.); f.khan@dundee.ac.uk (F.K.); i.mordi@dundee.ac.uk (I.R.M.)

**Keywords:** heart failure with reduced ejection fraction, outpatient, prognosis, renal dysfunction, blood urea nitrogen creatinine ratio, neurohormonal activation

## Abstract

Background: Chronic heart failure with reduced ejection fraction (HFrEF) is associated with high mortality, and renal dysfunction is common in these patients. Blood urea/creatinine ratio (UCR) has been identified as a potential prognostic marker, reflecting both renal function and neurohormonal activity. We assessed whether a UCR ≥ 95 at discharge from an outpatient service was associated with increased mortality. Methods: This retrospective study reviewed 337 patients (age 72.7 ± 14.3 years; 64.7% Male; Mean LVEF 33.2 ± 8.9%) with HFrEF referred to the Heart Failure Nurse Service at NHS Tayside for optimisation of heart failure medication. Cox proportional hazards models were used to assess the association between UCR and all-cause mortality. Results: Receiver operating characteristic (ROC) analysis identified a UCR threshold of 95 (area under the curve [AUC] 0.701) as predictive of mortality. Results demonstrated that a UCR ≥ 95 was independently associated with increased mortality (HR 1.85, 95% CI 1.09–3.14, *p* = 0.022). A high UCR was associated with increased mortality even in patients with preserved eGFR, a group typically considered at lower risk (HR 4.03, 95% CI 1.50–10.9, *p* = 0.006). Conclusions: These findings suggest that UCR could be a useful addition for identifying high-risk patients who may benefit from closer monitoring and more aggressive intervention following optimisation of heart failure medication.

## 1. Introduction

Chronic heart failure is a public health problem accounting for significant morbidity and mortality worldwide [1,2,3]. Direct and indirect costs of heart failure are significant, accounting for approximately 1% GDP in very high-human development index (HDI) countries but accounting for approximately 8% in low HDI countries [4]. Heart failure is often associated with renal dysfunction complicating the management as it is a limiting factor in initiating and titrating disease modifying agents such as ACE inhibitors and aldosterone antagonists [5]. Heart failure with renal dysfunction is defined as cardiorenal syndrome, with overlapping pathogenesis comprising neurohormonal activation, haemodynamic compromise and inflammation leading to fibrosis [6]. The recent ESC position statement regarding worsening renal failure recommends the use of blood urea monitoring in patients with heart failure with reduced ejection fraction (HFrEF) [7]. Although both serum creatinine and blood urea are markers for renal dysfunction, blood urea levels are also affected by reabsorption in the renal tubules, a process modulated by renin–angiotensin–aldosterone activity, sympathetic nervous activity and arginine-vasopressin activity [8]. Vasopressin acts on the V2 receptor of the inner medullary collecting duct to increase urea reabsorption and copeptin, a marker of vasopressin activity, is a strong prognostic marker of mortality in HFrEF [9,10]. Thus, neurohormonal activity is reflected in the blood urea/creatinine ratio (UCR) which effectively ‘normalises’ blood urea for the degree of renal dysfunction, as measured by the creatinine level [11].

Previous studies have investigated the prognostic utility of UCR in patients with HFrEF with a focus largely on patients admitted to hospital with acute decompensated heart failure [12,13]. The use of UCR in patients with chronic HFrEF receiving intensive titration of guideline-directed medical therapy (GDMT) in the outpatient setting has not been studied. Therefore, this retrospective study aimed to explore the association between UCR and mortality in a real-world cohort and to generate hypotheses regarding its utility independent of eGFR.

## 2. Materials and Methods

A retrospective analysis of the electronic health records of all patients referred to the Heart Failure Nurse Service (HFNS) in NHS Tayside, Scotland, UK in 2022 with a population approximately 400,000. Records were reviewed in February 2024. Inclusion criteria consisted of patients with symptomatic heart failure NYHA I–IV with echocardiogram or cardiac MRI demonstrating HFrEF at time of referral to the service. Patients were either diagnosed more than 3 months prior to first referral or previously diagnosed and referred for further titration of GDMT. Patients were discharged from the service when maximally titrated on GDMT and euvolaemic. While prior studies have often utilised a UCR cut-off of 100, we sought to determine the optimal data-driven threshold for our specific cohort [12]. A ROC curve analysis was performed. UCR was calculated as follows: UCR = Blood Urea (mmol/L)/Creatinine (µmol/L) × 1000

Statistical analysis was conducted using R version 4.4.2. Values are presented as mean ± standard deviation. Proportions are presented as value (proportion%). Prior to analysis, multicollinearity among predictors was assessed using Variance Inflation Factors and some variables were combined such as ACEi, ARB and ARNi as they demonstrated high multicollinearity as expected.

A Cox proportional hazards regression was conducted to evaluate the association between demographic, clinical, and treatment-related factors with time to death from diagnosis to capture the entire patient journey. Univariate Cox regression analyses were first performed for each independent variable to identify potential statistical predictors. Variables that were likely significant (*p* < 0.2) in the univariate models were then considered for inclusion in the multivariate Cox regression model [14]. To avoid overfitting given the small cohort, variables were minimised to approximately one per 10 events. Proportional hazard assumptions were assessed using log minus log. No violations of proportional hazards were detected.

To assess the specific association between UCR and cardiovascular mortality, a competing risks analysis was performed. Cardiovascular death was treated as the event of interest, and non-cardiovascular death was treated as a competing event. A Fine–Gray subdistribution hazard model was used to estimate the hazard ratios for the association between UCR and the cumulative incidence of cardiovascular death.

To test for a potential interaction between UCR and baseline renal function, an interaction term between the eGFR group (≥60 vs. <60 mL/min/1.73 m^2^) and UCR was included in a further model. To aid in the interpretation of the main effects in this interaction model, UCR was centered by subtracting its mean value from each observation. Hazard ratios (HR) are presented as HR (95% confidence intervals). Comparisons were deemed to be statistically significant when *p* < 0.05 (α = 0.05).

By performing a ROC curve analysis on UCR at discharge to predict death within 12 months, we evaluated the diagnostic accuracy of UCR as a prognostic marker. The area under the curve (AUC) was 0.701, indicating modest discriminative ability. The ROC analysis identified an optimal cutoff of 95.9, which yielded a sensitivity of 66.8% and specificity of 69.5%. As a result, a cutoff of 95 was chosen to enhance clinical applicability, ensure ease of recall in a clinical setting, and maintain consistency with similar cut-offs used in prior literature.

Kaplan–Meier analyses were performed on all patients with available data for the variable of interest (*n* = 267), while multivariable Cox regression was performed on the subset of patients with complete data for all covariates in the model (*n* = 264).

This study conforms with the principles outlined in the Declaration of Helsinki. NHS Tayside Information Governance granted Caldicott approval (IGTCAL-2024-016) in accordance with UK data protection and Caldicott principles. All data were anonymised prior to analysis.

## 3. Results

### 3.1. Patient Characteristics

During the calendar year 2022, 348 patients were referred to the Heart Failure Nurse Service (HFNS). After a review of the electronic health records, 11 were excluded due to duplication and insufficient data of the minimum required, resulting in 337 patient records suitable for analysis. The baseline characteristics of the cohort, stratified by mortality status at the end of the follow-up period, are presented in Table 1.

Patients who died during follow-up were significantly older, had a higher prevalence of cerebrovascular disease, and presented with more advanced heart failure as indicated by a lower systolic blood pressure, and worse NYHA functional classification (*p* < 0.001). Laboratory findings at both diagnosis and discharge revealed significantly worse renal function in the deceased group, including higher urea and creatinine levels, and lower eGFR (*p* < 0.001). Notably, patients who survived were significantly more likely to be prescribed guideline-directed medical therapy, including ACEi/ARB/ARNI, MRA, and SGLT2i, and were more likely to be on four agents concurrently (*p* < 0.001).

### 3.2. Prognostic Value of UCR Independent of Renal Function

To investigate the interplay between UCR and renal function, a multivariable Cox regression model was constructed using a combined four-category variable based on eGFR (<60 or ≥60 mL/min/1.73 m^2^) and UCR (<95 or ≥95). The results are detailed in Table 2.

The reference group was Group A (eGFR ≥ 60 + UCR < 95). Compared to this reference, Group B (eGFR ≥ 60 + UCR ≥ 95) had a significantly increased risk of death (HR 4.03, 95% CI 1.50–10.86, *p* = 0.006). Patients in Group D, who had both an eGFR < 60 and a UCR ≥ 95, also had a significantly elevated risk of mortality (HR 2.85, 95% CI 1.16–7.01, *p* = 0.023). Notably, patients in Group C, who had an eGFR < 60 but a normal UCR (<95), did not have a statistically significant increase in mortality risk in the adjusted model (HR 2.11, 95% CI 0.84–5.30, *p* = 0.112).

Kaplan–Meier analysis demonstrated a clear gradient of risk, with Group A (eGFR ≥ 60 + UCR < 95) showing the highest survival probability throughout the follow-up period (Figure 1). Survival in Group C (eGFR < 60 + UCR < 95) was also favourable compared to groups with elevated UCR. For both Group A and Group C, the median survival time was not reached during the study period. In contrast, groups with elevated UCR had substantially lower median survival times: Group D (eGFR < 60 + UCR ≥ 95) had a median survival of 108.5 months, and Group B (eGFR ≥ 60 + UCR ≥ 95) had the lowest median survival at 93.2 months. These findings underscore that elevated UCR is associated with poorer survival regardless of eGFR status.

Pairwise contrasts of adjusted survival demonstrated that patients with eGFR ≥ 60 and UCR < 95 (Group A) had significantly better survival than those with eGFR ≥ 60 and UCR ≥ 95 (Group B) (log HR = −1.40, 95% CI: −2.69 to −0.10, *p* = 0.030). Other comparisons did not reach statistical significance, however the point estimate for mortality risk was higher in Group D compared to Group A (log HR = −1.05, *p* = 0.104). These findings suggest that elevated UCR may be an independent prognostic marker, even in patients with preserved kidney function.

### 3.3. Prognostic Value of UCR at Discharge

A second multivariable Cox proportional hazards regression analysis was performed to assess predictors of all-cause mortality. After adjusting for age, sex, comorbidities, and use of guideline-directed medical therapies, a UCR ≥ 95 at discharge was independently associated with a significantly increased risk of death (HR 1.85, 95% CI 1.09–3.14, *p* = 0.022). UCR at diagnosis was not predictive of mortality (HR 0.94, 95% CI 0.51–1.75, *p* = 0.851).

Other significant predictors of mortality included a history of cerebrovascular disease (HR 2.69). Conversely, the use of MRA (HR 0.28), SGLT2i (HR 0.51) were all associated with a significantly lower risk of death. However, ACEi/ARB/ARNI therapy (HR 0.73) was not significant.

Survival analysis using the Kaplan–Meier method confirmed these findings (Figure 2). The median survival was significantly shorter in patients with a UCR ≥ 95 compared to those with a UCR < 95 (104.3 months vs. 209.6 months; log-rank *p* < 0.0001).

### 3.4. UCR as a Continuous Predictor and Model Validation

Of the 264 patients with complete data, 68 deaths occurred during follow-up. In the multivariable Cox model adjusting for age, sex, cerebrovascular disease (CVA), and heart failure therapies, each 1-unit increase in centered UCR was associated with a 1% increase in mortality (HR 1.010; 95% CI 1.003–1.018; *p* = 0.006). The interaction between UCR and eGFR status (<60 vs. ≥60 mL/min/1.73 m^2^) was not statistically significant (*p* = 0.15), indicating that the effect of UCR on mortality did not differ by renal function category. The model demonstrated good discrimination (concordance = 0.775). Bootstrap analyses supported the robustness of the UCR effect, with 95% percentile confidence intervals for the log hazard ratio ranging from 0.002 to 0.022.

### 3.5. Association of UCR with Cardiovascular-Specific Mortality

In a Fine–Gray competing risks regression model evaluating cardiovascular (CV) death, with non-CV death treated as a competing event, higher UCR at discharge was independently associated with an increased cumulative incidence of CV death. Patients with UCR ≥ 95 had a 2.83-fold higher sub-distribution hazard of CV death compared with those with UCR < 95 (sHR 2.83, 95% CI 1.28–6.28, *p* = 0.010). Treatment with MRA and SGLT2 inhibitors were both associated with substantially lower hazards of CV death (MRA sHR 0.16, 95% CI 0.07–0.37, *p* < 0.001; SGLT2i sHR 0.48, 95% CI 0.23–1.00, *p* = 0.049). There were no significant associations with age (sHR 1.01, 95% CI 0.97–1.04, *p* = 0.70), male sex (sHR 0.76, 95% CI 0.38–1.53, *p* = 0.40), cerebrovascular disease (sHR 1.17, 95% CI 0.43–3.21, *p* = 0.80), or ACEi/ARB/ARNI therapy (sHR 0.80, 95% CI 0.37–1.73, *p* = 0.60).

### 3.6. Sensitivity Analysis

A sensitivity analysis was performed to assess the robustness of UCR as a predictor after adjusting for diuretic dose and other significant univariate predictors (Haemoglobin, LVEF, and moderate or greater valvular disease). In this expanded model (Table A1), Haemoglobin, LVEF, and valvular disease were not found to be significant independent predictors. Importantly, UCR at discharge remained independently associated with mortality when treated as a continuous variable (HR 1.01, 95% CI 1.00–1.02, *p* = 0.038). This finding confirms the robustness of our primary analysis and suggests the prognostic value of UCR is independent of these key confounders.

## 4. Discussion

This study demonstrates that an elevated blood urea/creatinine ratio ≥ 95 at discharge from a heart failure nurse-led service is independently associated with a significantly increased risk of mortality in patients with chronic HFrEF. This finding reinforces the utility of UCR as a prognostic biomarker, extending its relevance beyond the acute setting, where it has been previously described, to a stable, ambulatory population undergoing optimisation of GDMT [12,15]. These findings are supported by the recent analysis by Tolomeo et al. of 28,000 participants of clinical trials which reported a UCR greater than 100 was associated with increased risk of cardiovascular death or HF hospitalisation, although the authors do note the limited inclusion criteria of the included trials including baseline eGFR [16]. While the prognostic value of UCR has been well-described in the setting of acutely decompensated patients and more recently in heart failure with preserved ejection fraction (HFpEF), its utility in a stable, ambulatory HFrEF cohort undergoing GDMT optimisation has been less clear [13,17].

The prognostic value of UCR likely reflects more than just renal dysfunction. Urea reabsorption in the renal tubules is modulated by neurohormonal activity, including renin–angiotensin–aldosterone system stimulation, vasopressin secretion, and sympathetic tone [18]. Therefore, elevated UCR may reflect heightened neurohormonal activation even when serum creatinine levels are not significantly deranged. This is further demonstrated by the non-significance of eGFR at discharge and weak significance at diagnosis. In this context, UCR may act as a surrogate for the systemic disease burden, offering prognostic insights independent of eGFR and creatinine. While the discriminative ability of UCR alone was modest, its primary clinical utility is its ability to stratify risk within established eGFR categories. Our primary finding is that UCR offers significant prognostic information that is not captured by eGFR alone rather than a comprehensive standalone tool.

The prognostic value of UCR remained significant even after accounting for diuretic dosage in a sensitivity analysis, further supporting its utility as a useful marker beyond simple haemoconcentration. Elevated UCR has been associated with venous congestion in heart failure. In a cohort of 103 outpatients with chronic HF and renal dysfunction, Parrinello et al. demonstrated that higher UCR correlated with inferior vena cava dilation and reduced collapsibility, both markers of elevated central venous pressure [19]. Raised UCR was also independently predictive of mortality over 31 months. The authors suggest that UCR may serve as a surrogate for residual congestion. In contrast, our cohort consisted of patients optimised on GDMT and clinically euvolemic at discharge, indicating that the prognostic value of UCR in this context may reflect persistent neurohormonal activation or subclinical cardiorenal stress rather than overt volume overload. To determine if this association was specific to the underlying disease process, we performed a competing risks analysis. The finding that UCR remained significantly associated with cardiovascular-specific death strengthens our conclusion that UCR is not merely a marker of general frailty but is intrinsically linked to the pathophysiology of HFrEF and adverse cardiorenal interactions.

Clinically, UCR represents a simple, inexpensive, and readily available marker that could be incorporated into clinical practice to identify high-risk patients who may benefit from closer monitoring or more aggressive therapeutic intervention. This is particularly relevant as high-risk HF patient groups have been shown to benefit from intensive follow-up strategies, such as telemonitoring [20,21]. Given the strong association with mortality, a UCR ≥95 could be used to highlight patients for more frequent renal function monitoring, or early consideration of advanced heart failure therapies such as finerenone. This aligns with a broader strategy of using readily available biomarkers to stratify risk and guide therapeutic intensity [22]. Notably, in our fully adjusted model, the group with isolated renal dysfunction (eGFR < 60) but a normal UCR did not have a statistically significant increase in mortality compared to the reference group (HR 2.11, 95% CI 0.84–5.30, *p* = 0.112). While this finding may reflect limited statistical power, it contrasts sharply to the highly significant, over fourfold risk observed in patients with a high UCR, even when eGFR was preserved.

This finding was further strengthened when UCR was analysed as a continuous variable, which showed a significant, graded increase in risk with each unit increase in UCR. Furthermore, a formal test for interaction confirmed that this effect was consistent regardless of whether patients had preserved or impaired renal function, underscoring that UCR offers prognostic information beyond a simple measure of kidney health. The overall model showed good discrimination (C-statistic 0.775), and its robustness was confirmed with internal validation. This suggests that the prognostic information provided by UCR may be more potent than that of eGFR in this context.

The discriminative performance of UCR was comparable to that of other biomarkers, including NT-proBNP. Noveanu et al. reported that NT-proBNP is a prognostic marker of mortality following hospital admission for acute heart failure (AUC 0.77) [23]. Additionally, a recent study investigating the serum creatinine/albumin ratio reported an AUC of 0.77 for predicting 28-day mortality following acute HF hospitalisation [24]. One study examined a machine learning score utilising multiple markers, MARKER-HF, which demonstrated greater discrimination at determining prognosis (AUC 0.83–0.89) compared to standalone biomarkers [25]. These comparisons highlight that while UCR demonstrates prognostic utility, it should be viewed as complementary to established biomarkers rather than as a replacement. Integrated multiple biomarker approaches are likely to provide the most accurate assessment of risk in patients with HFrEF.

This study’s strengths include its real-world cohort, application of survival and multivariable analyses, and rarely used but potentially modifiable marker. This study has several limitations. The retrospective design limits causal inference, and the modest sample size increases the risk of residual confounding, model overfitting and the potential for underpowered conclusions including subgroup analyses. A key limitation is the potential for unmeasured confounding. Factors such as patient frailty, detailed nutritional status or comorbid malignancy were not available for analysis. Although NT-proBNP was also unavailable, previous studies have shown that UCR and natriuretic peptides capture distinct pathophysiological pathways and provide complementary rather than overlapping prognostic information [26]. Therefore, the absence of NT-proBNP data is unlikely to explain the observed associations between UCR and outcomes. A further limitation is the potential for immortal time bias as patients had to survive from diagnosis to referral and through medicines titration prior to discharge from the service. Therefore, while our model adjusts for key covariates, we cannot definitively rule out that UCR acts as a surrogate for these unmeasured factors.

Future studies should seek to validate UCR as a prognostic and therapeutic stratification tool in prospective outpatient cohorts and compare with validated markers of prognostication including NT-proBNP. Randomised or implementation studies could explore whether targeting advanced therapies or closer follow-up in patients with elevated UCR improves clinical outcomes. The integration of UCR into decision support tools, interpreted alongside established biomarkers and clinical indicators, may ultimately offer a practical and scalable means to optimise heart failure care.

## 5. Conclusions

In this retrospective analysis, a UCR equal to or greater than 95 at discharge following guideline-directed medical therapy titration was independently associated with an increased mortality in patients with HFrEF. UCR may, therefore, serve as a practical marker for risk stratification in ambulatory HFrEF patients, though these results require validation in larger, prospective studies.

## Figures and Tables

**Figure 1 diseases-13-00362-f001:**
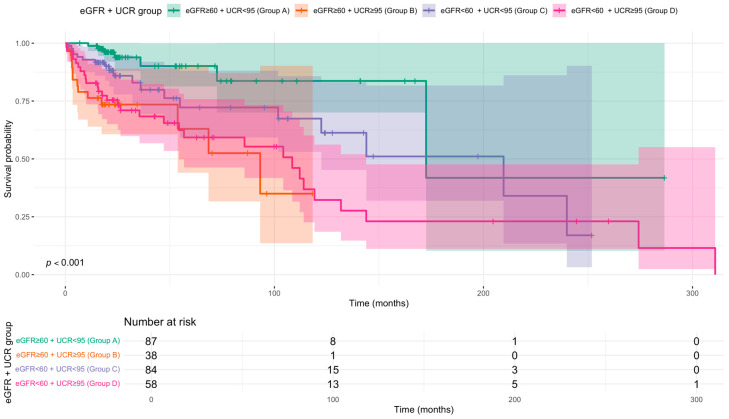
Kaplan–Meier survival curves for all-cause mortality stratified by discharge eGFR and UCR status. Patients were divided into four groups based on estimated glomerular filtration rate (eGFR < 60 vs. ≥60 mL/min/1.73 m^2^) and blood urea/creatinine ratio (UCR < 95 vs. ≥95). The *p*-value from the log-rank test is shown. Time since diagnosis in months. Shaded areas indicate 95% confidence intervals. Number at risk tabulated below.

**Figure 2 diseases-13-00362-f002:**
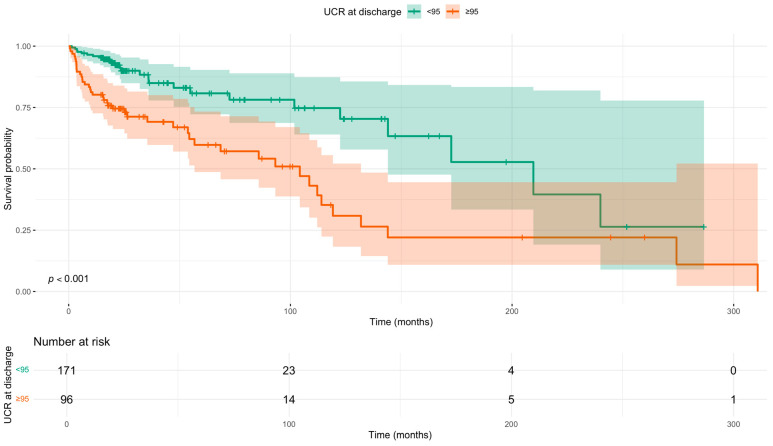
Kaplan–Meier survival curves for all-cause mortality stratified by discharge UCR (<95 vs. ≥95) status. *p*-value calculated using the log-rank test. Time since diagnosis in months. Shaded areas indicate 95% confidence intervals. Number at risk tabulated below.

**Table 1 diseases-13-00362-t001:** Baseline characteristics by mortality.

Baseline Characteristics by Mortality			
Variable	Alive ^1^	Dead ^1^	*p*-Value ^2^
Sex			0.6
Female	90 (34%)	29 (38%)	
Male	171 (66%)	47 (62%)	
Age (years)	71 (15)	79 (11)	<0.001
Cardiovascular Death		40 (53%)	
Cancer Death		7 (9%)	
Non-Cardiovascular or Cancer Death		29 (38%)	
BMI (kg/m^2^)	28 (7)	26 (7)	0.010
Atrial Fibrillation	112 (43%)	29 (39%)	0.5
Ischemic Heart Disease	108 (42%)	35 (47%)	0.5
Cerebrovascular Disease	29 (11%)	19 (25%)	0.004
Chronic Obstructive Pulmonary Disease	44 (17%)	17 (23%)	0.3
Asthma	35 (13%)	4 (5.4%)	0.065
Smoker (current or former)	78 (30%)	30 (40%)	0.12
Left Ventricular Ejection Fraction (%) ^3^	34 (9)	31 (8)	0.062
Left Ventricular Ejection Fraction Group ^3^			0.009
≥60%	3 (1.1%)	0 (0%)	
50–59%	14 (5.4%)	0 (0%)	
40–49%	45 (17%)	15 (20%)	
30–39%	118 (45%)	24 (32%)	
<30%	81 (31%)	37 (49%)	
Aortic Regurgitation ^4^			<0.001
None	199 (77%)	42 (57%)	
Mild	43 (17%)	29 (39%)	
Moderate	17 (6.6%)	3 (4.1%)	
Moderate–Severe	0 (0%)	0 (0%)	
Severe	0 (0%)	0 (0%)	
Tricuspid Regurgitation ^4^			0.13
None	105 (41%)	28 (38%)	
Mild	120 (47%)	31 (42%)	
Moderate	25 (9.8%)	10 (14%)	
Moderate–Severe	1 (0.4%)	3 (4.1%)	
Severe	5 (2.0%)	1 (1.4%)	
Pulmonary Regurgitation ^4^			0.065
None	145 (69%)	31 (54%)	
Mild	61 (29%)	26 (46%)	
Moderate	4 (1.9%)	0 (0%)	
Moderate–Severe	0 (0%)	0 (0%)	
Severe	0 (0%)	0 (0%)	
Mitral Stenosis ^4^			0.011
None	260 (100%)	73 (96%)	
Mild	0 (0%)	3 (3.9%)	
Moderate	0 (0%)	0 (0%)	
Moderate–Severe	0 (0%)	0 (0%)	
Severe	0 (0%)	0 (0%)	
Aortic Stenosis ^4^			0.2
None	233 (90%)	62 (83%)	
Mild	18 (6.9%)	7 (9.3%)	
Moderate	6 (2.3%)	4 (5.3%)	
Moderate–Severe	1 (0.4%)	0 (0%)	
Severe	2 (0.8%)	2 (2.7%)	
Tricuspid Stenosis ^4^			>0.9
None	252 (100%)	74 (100%)	
Mild	0 (0%)	0 (0%)	
Moderate	0 (0%)	0 (0%)	
Moderate–Severe	0 (0%)	0 (0%)	
Severe	0 (0%)	0 (0%)	
Pulmonary Stenosis ^4^			>0.9
None	229 (100%)	60 (100%)	
Mild	0 (0%)	0 (0%)	
Moderate	0 (0%)	0 (0%)	
Moderate–Severe	0 (0%)	0 (0%)	
Severe	0 (0%)	0 (0%)	
Systolic BP (mmHg)	121 (18)	115 (18)	0.032
Haemoglobin (g/L)	140 (18)	126 (25)	<0.001
Urea (diagnosis) (mmol/L)	7.8 (3.6)	9.9 (7.1)	0.049
Creatinine (diagnosis) (µmol/L)	93 (31)	112 (66)	0.019
eGFR (diagnosis) (mL/min/1.73 m^2^)			0.005
<15	0 (0%)	1 (1.3%)	
15–29	6 (2.3%)	7 (9.3%)	
30–44	26 (10%)	12 (16%)	
45–59	44 (17%)	14 (19%)	
≥60	183 (71%)	41 (55%)	
UCR (diagnosis)	84 (28)	87 (31)	0.3
Potassium (diagnosis) (mmol/L)	4.31 (0.52)	4.20 (0.49)	0.2
Sodium (diagnosis) (mmol/L)	138.8 (8.4)	138.1 (4.3)	0.057
Urea (discharge) (mmol/L)	9 (7)	19 (31)	<0.001
Creatinine (discharge) (µmol/L)	110 (42)	164 (130)	<0.001
UCR (discharge)	83 (26)	106 (37)	<0.001
eGFR (discharge) (mL/min/1.73 m^2^)			<0.001
<15	0 (0%)	6 (7.9%)	
15–29	17 (6.5%)	13 (17%)	
30–44	47 (18%)	20 (26%)	
45–59	61 (23%)	16 (21%)	
≥60	135 (52%)	21 (28%)	
Sodium (discharge) (mmol/L)	139.3 (2.8)	138.0 (7.6)	0.067
Potassium (discharge) (mmol/L)	4.53 (0.45)	4.47 (0.77)	0.11
NYHA Functional Classification			<0.001
1	56 (22%)	4 (5.8%)	
2	157 (62%)	29 (42%)	
3	34 (13%)	29 (42%)	
4	5 (2.0%)	7 (10%)	
GDMT agents			<0.001
1 agent	6 (2.3%)	18 (24%)	
2 agents	53 (21%)	22 (29%)	
3 agents	84 (33%)	26 (34%)	
4 agents	114 (44%)	10 (13%)	
Beta-blocker	221 (85%)	64 (84%)	>0.9
MRA	182 (70%)	29 (38%)	<0.001
SGLT2i	192 (74%)	39 (51%)	<0.001
ACEi/ARB/ARNI	225 (86%)	48 (63%)	<0.001
Deprivation Decile (Scottish Index of Multiple Deprivation)	10.5 (5.5)	9.8 (5.2)	0.3
Diuretic (Furosemide (mg))	30 (32)	57 (50)	<0.001

^1^ Mean (SD) or n (%); ^2^ Fisher’s exact test or Wilcoxon rank sum test; ^3^ Left Ventricular Ejection Fraction was measured at the most recent assessment prior to or at discharge from the service; ^4^ where valves were not visualised on echocardiography, these have been excluded.

**Table 2 diseases-13-00362-t002:** Multivariable cox regression analysis for all-cause mortality by eGFR and UCR Group.

Multivariable Cox Regression Analysis for All-Cause Mortality by eGFR and UCR Group			
Variable	Hazard Ratio	95% CI	*p*-Value
Age (years)	1.00	0.97, 1.02	0.7
Male	0.76	0.45, 1.26	0.3
Cerebrovascular Disease	2.68	1.49, 4.83	0.001
ACEi/ARB/ARNI	0.77	0.45, 1.33	0.4
SGLT2i	0.49	0.29, 0.83	0.007
MRA	0.27	0.15, 0.46	<0.001
eGFR + UCR group			
eGFR ≥ 60 + UCR < 95 (Group A)	–	–	
eGFR ≥ 60 + UCR ≥ 95 (Group B)	4.03	1.50, 10.9	0.006
eGFR < 60 + UCR < 95 (Group C)	2.11	0.84, 5.30	0.112
eGFR < 60 + UCR ≥ 95 (Group D)	2.85	1.16, 7.01	0.023

## Data Availability

The data presented in this study are available on request from the corresponding author due to patient confidentiality.

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
