# Peer review of "Blood Urea/Creatinine Ratio and Mortality in Ambulatory Patients with Heart Failure with Reduced Ejection Fraction"

_diseases, 2025, doi:10.3390/diseases13110362_

Round 1

Reviewer 1 Report

Comments and Suggestions for Authors

This retrospective analysis - according to line 50/51 it was intended to generate hypotheses - is based on 337 out of 348 patients with "chronic" (see line 47) HFrEF. But in line 57 the HF was either recently diagnosed or previously diagnosed and referred for further titration - that means the population is not consistent and the 2 groups had to be analyzed separately. The numbers of patients are not consistent: in Fig. 1 n=267 (plus the 76 deaths: 343 - in contrast to 337 of the abstract), in line 177 264 patients with complete data. These inconsistencies had to be corrected. The follow-up period of nearly 2 years (24 months) does not correspond i.e. to the period of up to 300 months in fig. 1 and 2. As mentioned in limitations unmeasured confounding factors could not be included, in addition to the named factors cancer was not mentioned, but may be of influence on mortality independent from UCR.         

Author Response

Dear Reviewer,

Thank you for taking the time to review our manuscript and for your constructive comments. We greatly appreciate your feedback which has helped us improve the clarity and quality of our work. Please see our point by point responses below.

Comment 1 - This retrospective analysis - according to line 50/51 it was intended to generate hypotheses - is based on 337 out of 348 patients with "chronic" (see line 47) HFrEF. But in line 57 the HF was either recently diagnosed or previously diagnosed and referred for further titration - that means the population is not consistent and the 2 groups had to be analyzed separately. 

Response 1 – We agree this is not clear; accordingly, we have clarified the language used. We have explicitly described that those recently diagnosed were at least 3 months post diagnosis by echo/CMR which is broadly defined as chronic HFrEF. We have updated this in the text line 57. 

Comment 2 - The numbers of patients are not consistent: in Fig. 1 n=267 (plus the 76 deaths: 343 - in contrast to 337 of the abstract), in line 177 264 patients with complete data. These inconsistencies had to be corrected.  

Response 2 – Thank you for highlighting this inconsistency. To rectify this, we have updated the methods to provide detail on why patient numbers vary according to the data available. Line 103/104. The n=267 in Figure 1 represents the total number of patients included in that specific survival analysis at time zero, a cohort which includes those who subsequently died during the follow-up period. 

Comment 3 - The follow-up period of nearly 2 years (24 months) does not correspond i.e. to the period of up to 300 months in fig. 1 and 2. 

Response 3 – We retrospectively analysed patients 2 years following referral to the service, however, to capture the entire disease trajectory we analysed our patients journey from diagnosis. We felt it was important to include this time as it an essential part of a patient’s journey with the disease doing so from clinic discharge would obscure this long-term prognosis. This is covered in lines 70/71. We acknowledge there is potential for immortal time bias and have updated our limitations to include this in line 284-286. 

Comment 4 - As mentioned in limitations unmeasured confounding factors could not be included, in addition to the named factors cancer was not mentioned, but may be of influence on mortality independent from UCR. 

Response 4 – Thank you for pointing this out. Unfortunately, we cannot confirm if any other patients had underlying malignancy and have updated the limitations to include this. We have been able to establish causes of death and can confirm 7 patients died of malignancy during follow up. Subsequently, we have been able to strengthen the results with a Fine Gray model demonstrating a greater HR when UCR>95 for cardiovascular mortality.  We hope this addition strengthens the findings and specificity. 

Reviewer 2 Report

Comments and Suggestions for Authors

This manuscript addresses the prognostic significance of the blood urea/creatinine ratio in patients with chronic heart failure with reduced ejection fraction undergoing optimisation of guideline-directed medical therapy in the outpatient setting. The topic is clinically relevant, and the dataset is valuable. The study is well written and the results are clearly presented. However, there are several important areas that require clarification or expansion before the manuscript can be considered for publication.

Major Comments.

-Valvular Heart Disease.

The authors do not report the prevalence of significant valvular heart disease (e.g., moderate/severe mitral regurgitation, aortic stenosis/regurgitation) in their cohort.

Given the close association between valvular pathology and prognosis in HFrEF, this is a major limitation. At minimum, baseline prevalence should be reported, and if data are available, adjustment for significant valvular disease should be considered in multivariable analysis.

-Severity of Left Ventricular Dysfunction.

The study reports mean LVEF but does not stratify patients by categories (e.g., mildly reduced [40–49%], moderately reduced [30–39%], severely reduced <30%).

Such stratification would provide additional clinical context, as prognosis differs markedly across these subgroups. At least descriptive data on how many patients fell into each category should be included. Multivariate analysis will be helpful.

-Diastolic Function Parameters

The association between UCR and diastolic function is not explored. Parameters such as E/e’, left atrial size, or filling pressures could have provided valuable mechanistic insights. Even if such data are not available, the limitation should be explicitly acknowledged.

-Limited References

The reference list is very short and omits much of the recent and relevant literature on renal biomarkers and heart failure prognosis.

The Discussion would be strengthened by a more comprehensive contextualisation of findings within the broader literature.

Minor Comments

-Statistical Methods

Data about valvular heart Valvular disease, EF and diastolic function should be included into multivariate analysis.

-Clinical Implications

The Discussion could be expanded to better explain how UCR measurement could be integrated into routine practice. For example, should it be considered alongside natriuretic peptides or troponins in prognostic assessment?

-Language and Style

The manuscript is generally well written but would benefit from some minor language editing to improve flow and reduce redundancy.

Author Response

Dear Reviewer, 

Thank you for taking the time to review our manuscript and for your constructive comments. We greatly appreciate your feedback which has helped us improve the clarity and quality of our work. Please see our point by point responses below. 

Comment 1 - Valvular Heart Disease. 

The authors do not report the prevalence of significant valvular heart disease (e.g., moderate/severe mitral regurgitation, aortic stenosis/regurgitation) in their cohort. 

Given the close association between valvular pathology and prognosis in HFrEF, this is a major limitation. At minimum, baseline prevalence should be reported, and if data are available, adjustment for significant valvular disease should be considered in multivariable analysis. 

Response 1 – Thank you for pointing this out. Regrettably, this information was not collected and is not available for analysis. We have updated the limitations to include this. Line 286/287. We hope this could be included in future prospective studies. 

Comment 2 - Severity of Left Ventricular Dysfunction. 

The study reports mean LVEF but does not stratify patients by categories (e.g., mildly reduced [40–49%], moderately reduced [30–39%], severely reduced <30%). 

Such stratification would provide additional clinical context, as prognosis differs markedly across these subgroups. At least descriptive data on how many patients fell into each category should be included. Multivariate analysis will be helpful. 

Response 2 – We agree this is a helpful clarification. We have updated the baseline characteristics to include the LVEF on the most recent echo/CMR. Notably, this does demonstrate some patients with LVEF>50% which demonstrates patient improvement on GDMT. Unfortunately, when categorised as recommended the model becomes highly unstable, likely due to few patients within some groups. When analysed as a continuous variable, it was not found to be significant predictor of mortality within this study and given the limited events, it was crucial to ensure the parsimonious model was as accurate as possible to prevent overfitting. 

Comment 3 -Diastolic Function Parameters 

The association between UCR and diastolic function is not explored. Parameters such as E/e’, left atrial size, or filling pressures could have provided valuable mechanistic insights. Even if such data are not available, the limitation should be explicitly acknowledged. 

Response 3 - Thank you for pointing this out. Regrettably, this information was not collected and is not available for analysis. We have updated the limitations to include this. Line 286/287. 

Comment 4 - Limited References 

The reference list is very short and omits much of the recent and relevant literature on renal biomarkers and heart failure prognosis. 

The Discussion would be strengthened by a more comprehensive contextualisation of findings within the broader literature. 

Response 4 – Thank you for emphasising this. We have since revisited the literature and updated this as highlighted. Lines 264-275 

Comment 5 - -Statistical Methods 

Data about valvular heart Valvular disease, EF and diastolic function should be included into multivariate analysis. 

Response 5 – As addressed above.  

Comment 6 - -Clinical Implications 

The Discussion could be expanded to better explain how UCR measurement could be integrated into routine practice. For example, should it be considered alongside natriuretic peptides or troponins in prognostic assessment? 

Response 6 – We agree and have expanded how we envision UCR could be integrated into clinical practice alongside other biomarkers following further validation. Lines 271-275, 292, 295. 

Comment 7 -Language and Style 

The manuscript is generally well written but would benefit from some minor language editing to improve flow and reduce redundancy. 

Response 7 – We have reviewed the text for redundancy and made the relevant changes. 

Reviewer 3 Report

Comments and Suggestions for Authors

I have enjoyed reading you manuscript entitled “Blood Urea/Creatinine Ratio and Mortality in Ambulatory 2 Patients with Heart Failure with Reduced Ejection Fraction”. The manuscript is well written, methodologically sound, and presents findings that may be clinically relevant. However, I have a few suggestions and comments that may help to strengthen your manuscript as follows:

  1. The sample size was relatively small for subgroup analyses, and this may be underpowered for detecting certain differences. This should be discussed as a limitation.
  2. Other important covariates such as NT-proBNP levels, frailty indices, or nutritional status are known to influence the major events in HFrEF. However, in this study, these measurements were not available. The authors should discuss how these unmeasured factors might influence the observed associations.
  1. The study highlights that UCR ≥95 is considered as a risk marker. How this could influence outpatient management strategies.
  2. The AUC for UCR is considered modest (0.701). The authors need to clarify how this level of discrimination compares to existing markers and the added value of UCR when combined with other clinical predictors.
  3. The authors used alternative terminology throughout the manuscript (e.g., “HFrEF” vs “chronic heart failure with reduced ejection fraction”). They need to be consistent.
  4. The conclusion could be made slightly more concise while emphasising the need for prospective validation.

Best regards,

Author Response

Dear Reviewer, 

Thank you for taking the time to review our manuscript and for your constructive comments. We greatly appreciate your feedback which has helped us improve the clarity and quality of our work. Please see our point by point responses below. 

Comment 1 - The sample size was relatively small for subgroup analyses, and this may be underpowered for detecting certain differences. This should be discussed as a limitation. 

Response 1 – Thank you, we have included this in our limitations. Line 280. 

Comment 2 - Other important covariates such as NT-proBNP levels, frailty indices, or nutritional status are known to influence the major events in HFrEF. However, in this study, these measurements were not available. The authors should discuss how these unmeasured factors might influence the observed associations. 

Response 2 – We agree. We have therefore updated the limitations to discuss these further. Line 283-284 

Comment 3 - The study highlights that UCR ≥95 is considered as a risk marker. How this could influence outpatient management strategies. 

Response 3 – Thank you we have suggested this could guide further management including closer follow-up alongside consideration of further therapies including finerenone. Line 248-250, 292-294 

Comment 4 - The AUC for UCR is considered modest (0.701). The authors need to clarify how this level of discrimination compares to existing markers and the added value of UCR when combined with other clinical predictors. 

Response 4 - Thank you, we have included other markers for comparison and are happy to report similar AUC with single biomarkers. MARKER-HF reported greater discrimination. Lines 264-275 

Comment 5 - The authors used alternative terminology throughout the manuscript (e.g., “HFrEF” vs “chronic heart failure with reduced ejection fraction”). They need to be consistent. 

Response 5 - We agree. We have reviewed the manuscript and standardized the terminology (e.g., consistently using 'HFrEF' or 'chronic HFrEF') throughout for clarity and consistency. 

Comment 6 - The conclusion could be made slightly more concise while emphasising the need for prospective validation. 

Response 6 - We agree and have revised this. Lines 300/301 

Round 2

Reviewer 1 Report

Comments and Suggestions for Authors

Te revised version still contains some serious flaws, which have to be corrected:

  • the multivariable regression analysis does not contain all statistically significant different parameters like LVEF Group or hemoglobin, in addition the parameters used for the analysis should be declared and reasons for not inclusion should be explained;
  • why sensitivity of 66.8% and specificity of 69.5% are accetable should be discussed / justified;
  • line 166/7: a "notable trend" without statistical difference in a scientific paper is inadequate;
  • as mentioned in discussion there are some serious limitations (i.e. unmeasured confounding factors etc.) which do nor allow the statement (line 295/6) that UCR is a significant predictor of increased mortality - this statement does not correspond to line 291/2 "...may offer ...". 

Author Response

Dear Reviewer, 

Thank you for taking the time to review our manuscript and for your constructive comments. We greatly appreciate your ongoing feedback which has helped us improve the clarity and quality of our work. Please see our point by point responses below. 

Comment 1 - the multivariable regression analysis does not contain all statistically significant different parameters like LVEF Group or hemoglobin, in addition the parameters used for the analysis should be declared and reasons for not inclusion should be explained; 

Response 1 - We thank the reviewer for this point. We have comprehensively addressed this in two ways in the edited manuscript: 

1. Justification for the parsimonious model: In the Methods (Line 80-81), we state: 'Given the small cohort to avoid overfitting, variables were minimised to approximately one per 10 events.' This is the standard statistical justification for not including all significant univariate predictors in the main model. 

2. Dedicated Sensitivity Analysis: To directly address the concern of confounding from these variables, we performed a new, robust sensitivity analysis. This is presented in Section 3.6 (Lines 215-222) and detailed in Appendix A (Table A1). 

This sensitivity analysis includes LVEF, Haemoglobin, and valvular disease. The results, as stated in Lines 1195-1196, show that UCR remained independently associated with mortality (HR 1.01, 95% CI 1.00-1.02, p=0.038), confirming our primary finding is robust to these confounders.  

Comment 2 - why sensitivity of 66.8% and specificity of 69.5% are accetable should be discussed / justified; 

Response 2 – This is an important point and is explicitly discussed in the Discussion section (Lines 244 -248). We acknowledge these values are modest and state: 

“While the discriminative ability of UCR alone was modest, its true clinical utility appears to lie in its ability to stratify risk within established eGFR categories. Our primary finding is not that UCR is a perfect screening tool, but that it offers significant prognostic information that is not captured by eGFR alone.” 

We believe this paragraph correctly frames the UCR, not as a standalone diagnostic test, but as a widely available risk stratification tool that adds prognostic value. 

Comment 3 - line 166/7: a "notable trend" without statistical difference in a scientific paper is inadequate; 

Response 3 – We completely agree. This wording has been corrected in the current draft. The revised text in Line 167/168 now reads: “Other comparisons did not reach statistical significance, however the point estimate for mortality risk was higher in Group D compared to Group A (log HR = -1.05, p=0.104).” This new wording is objective and states only the statistical facts. 

Comment 4 - as mentioned in discussion there are some serious limitations (i.e. unmeasured confounding factors etc.) which do nor allow the statement (line 295/6) that UCR is a significant predictor of increased mortality - this statement does not correspond to line 291/2 "...may offer ...". 

Response 4 - We agree with the reviewer that the original conclusion was too strong. This has been corrected in the current draft. The Conclusion section (Lines 321-325) has been entirely rewritten to be more cautious and to align perfectly with the limitations. We believe this revised wording accurately reflects the nature of our findings as a retrospective, hypothesis-generating study. 

Reviewer 2 Report

Comments and Suggestions for Authors

The manuscript has improved; however, several major issues remain unresolved, and a major revision is required.

The absence of data regarding valvular involvement in this cohort substantially limits the scientific strength of the study. I assume that all patients underwent formal echocardiographic evaluation and that the corresponding reports are available in the hospital’s medical archives. The authors are strongly encouraged to review these echocardiography reports and include information on significant valvular lesions—defined as moderate or greater regurgitation or stenosis. Furthermore, valvular involvement should be incorporated into the multivariate analysis, as it may represent an important confounding factor influencing the reported outcomes.

In addition, the reference list remains insufficient for a scientific article of this type. A more comprehensive review and integration of relevant existing literature will significantly enhance the depth and scholarly quality of the manuscript.

Author Response

Dear Reviewer, 

Thank you for taking the time to review our manuscript and for your constructive comments. We greatly appreciate your ongoing feedback which has helped us improve the clarity and quality of our work. Please see our point by point responses below. 

Comment 1 - The absence of data regarding valvular involvement in this cohort substantially limits the scientific strength of the study. I assume that all patients underwent formal echocardiographic evaluation and that the corresponding reports are available in the hospital’s medical archives. The authors are strongly encouraged to review these echocardiography reports and include information on significant valvular lesions—defined as moderate or greater regurgitation or stenosis. Furthermore, valvular involvement should be incorporated into the multivariate analysis, as it may represent an important confounding factor influencing the reported outcomes. 

Response 1 – We thank the reviewer for this valuable suggestion. We have revisited the echocardiographic data and extracted information on significant valvular lesions, defined as moderate or greater regurgitation or stenosis. These data have now been incorporated into the manuscript and summarised in Table 1. In addition, we have included valvular involvement in our multivariable Cox regression model as an independent covariate. Inclusion of this parameter did not materially alter the hazard ratio or significance of UCR, confirming the robustness of our findings. The updated analysis has been presented in the revised Appendix (Table A1), and corresponding details have been added to the Results and Methods sections. 

Comment 2 - In addition, the reference list remains insufficient for a scientific article of this type. A more comprehensive review and integration of relevant existing literature will significantly enhance the depth and scholarly quality of the manuscript. 

Responses 2 – Thank you, we have revisited the literature and expanded our introduction and discussion accordingly.  

Round 3

Reviewer 1 Report

Comments and Suggestions for Authors

The revised version is now acceptable.

Reviewer 2 Report

Comments and Suggestions for Authors

The manuscript has been substantially improved, all the issues have been resolved.